# Kernel Deformed Exponential Families for Sparse Continuous Attention

## Abstract

Attention mechanisms take an expectation of a data representation with respect to probability weights. This creates summary statistics that focus on important features. Recently, Martins et al. (2020; 2021) proposed continuous attention mechanisms, focusing on unimodal attention densities from the exponential and deformed exponential families: the latter has sparse support. Farinhas et al. (2021) extended this to use Gaussian mixture attention densities, which are a flexible class with dense support. In this paper, we extend this to two general flexible classes: kernel exponential families (Canu & Smola, 2006) and our new sparse counterpart kernel *deformed* exponential families. Theoretically, we show new existence results for both kernel exponential and deformed exponential families, and that the deformed case has similar approximation capabilities to kernel exponential families. Experiments show that kernel deformed exponential families can attend to multiple compact regions of the data domain.

## 1 Introduction

Attention mechanisms take weighted averages of data representations (Bahdanau et al., 2015), where the weights are a function of input objects. These are then used as inputs for prediction. Discrete attention 1) cannot easily handle data where observations are irregularly spaced 2) attention maps may be scattered, lacking focus. Martins et al. (2020; 2021) extended attention to continuous settings, showing that attention densities maximize the regularized expectation of a function of the data location (i.e. time). Special case solutions lead to exponential and deformed exponential families: the latter has sparse support. They form a continuous data representation and take expectations with respect to attention densities. Using measure theory to unify discrete and continuous approaches, they show transformer self-attention (Vaswani et al., 2017) is a special case of their formulation.

Martins et al. (2020; 2021) explored unimodal attention densities: these only give high importance to one region of data. Farinhas et al. (2021) extended this to use multi-modal mixture of Gaussian attention densities. However 1) mixture of Gaussians do not lie in either the exponential or deformed exponential families, and are difficult to study in the context of Martins et al. (2020; 2021) 2) they have dense support. Sparse support can say that certain regions of data do not matter: a region of time has *no* effect on class probabilities, or a region of an image is *not* some object. We would like to use multimodal exponential and deformed exponential family attention densities, and understand how Farinhas et al. (2021) relates to the optimization problem of Martins et al. (2020; 2021).

This paper makes three contributions: 1) we introduce kernel *deformed* exponential families, a multimodal class of densities with sparse support and apply it along with the multimodal kernel exponential families (Canu & Smola, 2006) to attention mechanisms. The latter have been used for density estimation, but not weighting data importance 2) we theoretically analyze normalization for both kernel exponential and deformed exponential families in terms of a base density and kernel, and show approximation properties for the latter 3) we apply them to real world datasets and show that kernel deformed exponential families learn flexible continuous attention densities with sparse support. Approximation properties for the kernel deformed are challenging: similar kernel exponential family results (Sriperumbudur et al., 2017) relied on standard exponential and logarithm properties to bound the difference of the log-partition functional at two functions: these do not hold for deformed analogues. We provide similar bounds via the functional mean value theorem along with bounding the Frechet derivative of the deformed log-partition functional.

The paper is organized as follows: we review continuous attention (Martins et al., 2020; 2021). We then describe how mixture of Gaussian attention densities, used in Farinhas et al. (2021), solve a different optimization problem. We next describe kernel exponential families and give novel normalization condition relating the kernel growth to the base density's tail decay. We then propose kernel deformed exponential families, which can have support over disjoint regions. We describe normalization and prove approximation capabilities. Next we describe use of these densities for continuous attention, including experiments where we show that the kernel deformed case learns multimodal attention densities with sparse support. We conclude with limitations and future work.

## 2 RELATED WORK

**Attention Mechanisms** closely related are Martins et al. (2020; 2021); Farinhas et al. (2021); Tsai et al. (2019); Shukla & Marlin (2021). Martins et al. (2020; 2021) frame continuous attention as an expectation of a value function over the domain with respect to a density, where the density solves an optimization problem. They only used unimodal exponential and deformed exponential family densities: we extend this to the multimodal setting by leveraging kernel exponential families and proposing a deformed counterpart. Farinhas et al. (2021) proposed a multi-modal continuous attention mechanism via a mixture of Gaussians approach. We show that this solves a slightly different optimization problem from Martins et al. (2020; 2021), and extend to two further general density classes. Shukla & Marlin (2021) provide an attention mechanism for irregularly sampled time series by use of a continuous-time kernel regression framework, but do not actually take an expectation of a data representation over time with respect to a continuous pdf, evaluating the kernel regression model at a fixed set of time points to obtain a discrete representation. This describes importance of data at a set of points rather than over regions. Other papers connect attention and kernels, but focus on the discrete attention setting (Tsai et al., 2019; Choromanski et al., 2020). Also relevant are temporal transformer papers, including Xu et al. (2019); Li et al. (2019; 2020); Song et al. (2018). However none have continuous attention densities.

**Kernel Exponential Families** Canu & Smola (2006) proposed kernel exponential families: Sriperumbudur et al. (2017) analyzed theory for density estimation. Wenliang et al. (2019) parametrized the kernel with a deep neural network. Other density estimation papers include Arbel & Gretton (2018); Dai et al. (2019); Sutherland et al. (2018). We apply kernel exponential families as attention densities to *weight* a value function which represents the data, rather than to estimate the data density, and extend similar ideas to kernel deformed exponential families with sparse support.

Wenliang et al. (2019) showed a condition for an unnormalized kernel exponential family density to have a finite normalizer. However, they used exponential power base densities. We instead relate kernel growth rates to the base density tail decay, allowing non-symmetric base densities.

To summarize our theoretical contributions: 1) introducing kernel *deformed* exponential families with approximation and normalization analysis 2) improved kernel exponential family normalization results.

## 3 CONTINUOUS ATTENTION MECHANISMS

An attention mechanism involves: 1) the value function approximates a data representation. This may be the original data or a learned representation. 2) the attention density is chosen to be 'similar' to another data representation, encoding it into a density 3) the context combines the two, taking an expectation of the value function with respect to the attention density. Formally, the context is

$$c = \mathbb{E}_{T \sim p}[V(T)]. \tag{1}$$

Here $V(t)$ the value function approximates a data representation, $T \sim p(t)$ is the random variable or vector for locations (temporal, spatial, etc), and $p(t)$ is the attention density.

To choose the attention density $p$, one takes a data representation $f$ and finds $p$ 'similar' to $f$ and thus to a data representation, but regularizing $p$. Martins et al. (2020; 2021) did this, providing a rigorous formulation of attention mechanisms. Given a probability space $(S, \mathcal{A}, Q)$, let $\mathcal{M}^1_+(S)$ be the set of densities with respect to $Q$. Assume that $Q$ is dominated by a measure $\nu$ (i.e. Lebesgue) and that it has density $q_0 = \frac{dQ}{d\nu}$ with respect to $\nu$. Let $S \subseteq \mathbb{R}^D$, $\mathcal{F}$ be a function class, and $\Omega : \mathcal{M}^1_+(S) \to \mathbb{R}$ be a lower semi-continuous, proper, strictly convex functional.

Given $f \in \mathcal{F}$, an *attention density* (Martins et al., 2020) $\hat{p} : \mathcal{F} \to \mathbb{R}_{\geq 0}$ solves

$$\hat{p}[f] = \arg \max_{p \in \mathcal{M}_+^1(S)} \langle p, f \rangle_{L^2(Q)} - \Omega(p). \tag{2}$$

This maximizes regularized $L^2$ similarity between $p$ and a data representation $f$. If $\Omega(p)$ is the negative differential entropy, the attention density is Boltzmann Gibbs

$$\hat{p}[f](t) = \exp(f(t) - A(f)), \tag{3}$$

where $A(f)$ ensures $\int_S \hat{p}[f](t)dQ = 1$. If $f(t) = \theta^T \phi(t)$ for parameters and statistics $\theta \in \mathbb{R}^M, \phi(t) \in \mathbb{R}^M$ respectively, Eqn. 3 becomes an exponential family density. For $f$ in a reproducing kernel Hilbert space $\mathcal{H}$, it becomes a kernel exponential family density (Canu & Smola, 2006), which we propose to use as an alternative attention density.

One desirable class would be heavy or thin tailed exponential family-like densities. In exponential families, the support, or non-negative region of the density, is controlled by the measure $Q$. Letting $\Omega(p)$ be the $\alpha$-Tsallis negative entropy $\Omega_\alpha(p)$ (Tsallis, 1988),

$$\Omega_\alpha(p) = \begin{cases} \frac{1}{\alpha(\alpha-1)} \left( \int_S p(t)^\alpha dQ - 1 \right), \alpha \neq 1; \\ \int_S p(t) \log p(t)dQ, \alpha = 1, \end{cases}$$

then $\hat{p}[f]$ for $f(t) = \theta^T \phi(t)$ lies in the deformed exponential family (Tsallis, 1988; Naudts, 2004)

$$\hat{p}_{\Omega_\alpha}[f](t) = \exp_{2-\alpha}(\theta^T \phi(t) - A_\alpha(f)), \tag{4}$$

where $A_\alpha(f)$ again ensures normalization and the density uses the $\beta$-exponential

$$\exp_\beta(t) = \begin{cases} [1 + (1 - \beta)t]_+^{1/(1-\beta)}, \beta \neq 1; \\ \exp(t), \beta = 1. \end{cases} \tag{5}$$

For $\beta < 1$, Eqn. 5 and thus deformed exponential family densities for $1 < \alpha \leq 2$ can return 0 values. Values $\alpha > 1$ (and thus $\beta < 1$) give thinner tails than the exponential family, while $\alpha < 1$ gives fatter tails. Setting $\beta = 0$ is called *sparsemax* (Martins & Astudillo, 2016). In this paper, we assume $1 < \alpha \leq 2$, which is the sparse case studied in Martins et al. (2020). We again propose to replace $f(t) = \theta^T \phi(t)$ with $f \in \mathcal{H}$, which leads to the novel *kernel deformed exponential families*.

Computing Eqn. 1's context vector requires parametrizing $V(t)$. Martins et al. (2020) obtain a value function $V : S \to \mathbb{R}^D$ parametrized by $\mathbf{B} \in \mathbb{R}^{D \times N}$ by applying regularized multivariate linear regression to estimate $V(t; \mathbf{B}) = \mathbf{B}\Psi(t)$, where $\Psi = \{\psi_n\}_{n=1}^N$ is a set of basis functions. Let $L$ be the number of observation locations (times in a temporal setting), $O$ be the observation dimension, and $N$ be the number of basis functions. This involves regressing the observation matrix $\mathbf{H} \in \mathbb{R}^{O \times L}$ on a matrix $\mathbf{F} \in \mathbb{R}^{N \times L}$ of basis functions $\{\psi_n\}_{n=1}^N$ evaluated at observation locations $\{t_l\}_{l=1}^L$

$$\mathbf{B}^* = \arg \min_{\mathbf{B}} \|\mathbf{B}\mathbf{F} - \mathbf{H}\|_F^2 + \lambda\|\mathbf{B}\|_F^2. \tag{6}$$

### 3.1 GAUSSIAN MIXTURE MODEL

Farinhas et al. (2021) used mixture of Gaussian attention densities, but did not relate this to the optimization definition of attention densities in Martins et al. (2020; 2021). In fact their attention densities solve a related but different optimization problem. Martins et al. (2020; 2021) show that exponential family attention densities maximize a regularized linear predictor of the expected sufficient statistics of locations. In contrast, Farinhas et al. (2021) find a joint density over locations and latent states, and maximize a regularized linear predictor of the expected joint sufficient statistics. They then take the marginal location densities to be the attention densities.

Let $\Omega(p)$ be Shannon entropy and consider two optimization problems:

$$\arg \max_{p \in \mathcal{M}_+^1(S)} \langle \theta, \mathbb{E}_p[\phi(T)] \rangle_{l^2} - \Omega(p)$$

$$\arg \max_{p \in \mathcal{M}_+^1(S)} \langle \theta, \mathbb{E}_p[\phi(T, Z)] \rangle_{l^2} - \Omega(p)$$

The first is Eqn. 2 with $f = \theta^T \phi(t)$ and rewritten to emphasize expected sufficient statistics. If one solves the second with variables $Z$, we recover an Exponential family joint density

$$\hat{p}_{\Omega_\alpha}[f](t, z) = \exp(\theta^T \phi(t, z) - A(\theta)).$$

This encourages the joint density of $T, Z$ to be similar to a *complete data* representation $\theta^T \phi(t, z)$ of both location variables $T$ and latent variables $Z$, instead of encouraging the density of $T$ to be similar to an observed data representation $\theta^T \phi(t)$. The latter optimization is equivalent to

$$\arg \max_{p \in \mathcal{M}_+^1(S)} \Omega(p)$$
$$\text{s.t.}$$
$$\mathbb{E}_{p(T,Z)}[\phi_m(T, Z)] = c_m, m = 1, \cdots, M.$$

The constraint terms $c_m$ are determined by $\theta$. Thus, this maximizes the joint entropy of $Z$ and $T$, subject to constraints on the expected joint sufficient statistics.

To recover EM learned Gaussian mixture densities, one must select $\phi_m$ so that the marginal distribution of $T$ will be a mixture of Gaussians, and relate $c_m$ to the EM algorithm used to learn the mixture model parameters. For the first, assume that $Z$ is a multinomial random variable taking $|Z|$ possible values and let $\phi(t, z) = (z_1, z_2, \cdots, z_{|Z|-1}, I(z = 1)t, I(z = 1)t^2, \cdots, I(z = |Z|)t, I(z = |Z|)t^2)$. These are multinomial sufficient statistics, followed by the sufficient statistics of $|Z|$ Gaussians multiplied by indicators for each $z$. Then $p(T|Z)$ will be Gaussian, $p(Z)$ will be multinomial, and $p(T)$ will be a Gaussian mixture. For contraints, Farinhas et al. (2021) have

$$\mathbb{E}_{p(T,Z)}[\phi_m(T, Z)] = \sum_{l=1}^{L} w_l \sum_{z=1}^{|Z|} p_{\text{old}}(z|t_l)\phi_m(t_l, z), m = 1, \cdots, M \tag{7}$$

at each EM iteration. Here $p_{\text{old}}(z|x_l)$ is the previous iteration's latent state density conditional on the observation value, $w_l$ are discrete attention weights, and $t_l$ is a discrete attention location. That EM has this constraint was shown in Wang et al. (2012). Intuitively, this matches the expected joint sufficient statistics to those implied by discrete attention over locations, taking into account the dependence between $z$ and $t_l$ given by old model parameters. An alternative is simply to let $\theta$ be the output of a neural network. While the constraints lack the intuition of Eqn. 7, it avoids the need to select an initialization. We focus on this case and use it for our baselines: both approaches are valid.

## 4 KERNEL EXPONENTIAL AND DEFORMED EXPONENTIAL FAMILIES

We now use kernel exponential families and a new deformed counterpart to obtain flexible attention densities solving Eqn. 2 with the same regularizers. We first review kernel exponential families. We then give a novel theoretical result describing when an unnormalized kernel exponential family density can be normalized. Next we introduce kernel deformed exponential families, extending kernel exponential families to have either sparse support or fatter tails: we focus on the former. These can attend to multiple non-overlapping time intervals or spatial regions. We show similar normalization results based on the choice of kernel and base density. Following this we show approximation theory. We conclude by showing how to compute attention densities in practice.

Kernel exponential families (Canu & Smola, 2006) extend exponential family distributions, replacing $f(t) = \theta^T \phi(t)$ with $f$ in a reproducing kernel Hilbert space $\mathcal{H}$ (Aronszajn, 1950) with kernel $k : S \times S \to \mathbb{R}$. Densities can be written as

$$p(t) = \exp(f(t) - A(f))$$
$$= \exp(\langle f, k(\cdot, t)\rangle_{\mathcal{H}} - A(f)),$$

where the second equality follows from the reproducing property. A challenge is to choose $\mathcal{H}, Q$ so that a normalizing constant exists, i.e., $\int_S \exp(f(t))dQ < \infty$. Kernel exponential family distributions can approximate any continuous density over a compact domain arbitrarily well in KL divergence, Hellinger, and $L^p$ distance (Sriperumbudur et al., 2017). However relevant integrals including the normalizing constant generally require numerical integration.

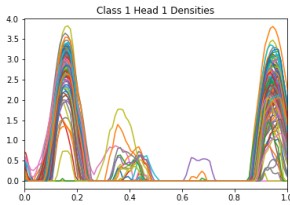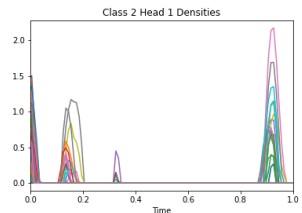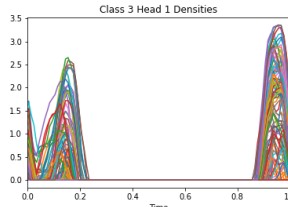

Figure 1: Attention densities for kernel deformed exponential families for the first attention head of the uWave experiment and all test set participants for three classes. The densities are sparse and each have support over different non-overlapping time intervals, which cannot be done with either Gaussian mixtures or exponential families. They also attend to similar regions within each class.

To avoid infinite dimensionality one generally assumes a representation of the form

$$f = \sum_{i=1}^{I} \gamma_i k(\cdot, t_i),$$

where for density estimation (Sriperumbudur et al., 2017) the $t_i$ are the observation locations. However, this requires using one parameter per observation value. This level of model complexity may not be necessary, and often one chooses a set of *inducing points* (Titsias, 2009) $\{t_i\}_{i=1}^{I}$ where $I$ is less than the number of observation locations.

For a given pair $\mathcal{H}, k$, how can we choose $Q$ to ensure that the normalization constant exists? We first give a simple example of $\mathcal{H}, f$ and $Q$ where the normalizing constant *does not* exist.

**Example 1.** *Let $Q$ be the law of a $\mathcal{N}(0, 1)$ distribution and $S = \mathbb{R}$. Let $\mathcal{H} = span\{t^3, t^4\}$ with $k(x, y) = x^3 y^3 + x^4 y^4$ and $f(t) = t^3 + t^4 = k(t, 1)$. Then*

$$\int_S \exp(f(t)) dQ = \int_{\mathbb{R}} \exp(\frac{t^2}{2} + t^3 + t^4) dt \tag{8}$$

*where the integral diverges.*

### 4.1 THEORY FOR KERNEL EXPONENTIAL FAMILIES

We provide sufficient conditions for $Q$ and $\mathcal{H}$ so that $A(f)$ the log-partition function exists. We relate $\mathcal{H}$'s kernel growth rate to the tail decay of the random variable or vector $T_Q$ with law $Q$.

**Proposition 1.** *Let $\tilde{p}(t) = \exp(f(t))$ where $f \in \mathcal{H}$ an RKHS with kernel $k$. Assume $k(t, t) \leq L_k \|t\|_2^{\xi} + C_k$ for constants $L_k, C_k, \xi > 0$. Let $Q$ be the law of a random vector $T_Q$, so that $Q(A) = P(T_Q \in A)$. Assume $\forall u$ s.t. $\|u\|_2 = 1$,*

$$P(|u^T T_Q| \geq z) \leq C_q \exp(-vz^{\eta}) \tag{9}$$

*for some constants $\eta > \frac{\xi}{2}, C_Q, v > 0$. Then*

$$\int_S \tilde{p}(t) dQ < \infty.$$

*Proof.* See Appendix A.1 □

Based on $k(t, t)$'s growth, we can vary what tail decay rate for $T_Q$ ensures we can normalize $\tilde{p}(t)$. Wenliang et al. (2019) also proved normalization conditions, but focused on random variables with exponential power density for a specific growth rate of $k(t, t)$ rather than relating tail decay to growth rate. By focusing on tail decay, our result can be applied to non-symmetric base densities. Specific kernel bound growth rate terms $\xi$ lead to allowing different tail decay rates.

**Corollary 1.** *For $\xi = 4$, $T_Q$ can be any sub-Gaussian random vector. For $\xi = 2$ it can be any sub-exponential. For $\xi = 0$ it can have any density.*

*Proof.* See Appendix A.2 □

### 4.2 KERNEL DEFORMED EXPONENTIAL FAMILIES

We now propose kernel deformed exponential families: flexible sparse non-parametric distributions. These take deformed exponential families and extend them to use kernels in the deformed exponential term. This mirrors kernel exponential families. We write

$$p(t) = \exp_{2-\alpha}(f(t) - A_\alpha(f)),$$

where $f \in \mathcal{H}$ with kernel $k$. Fig. 1b shows that they can have support over disjoint intervals.

#### 4.2.1 NORMALIZATION THEORY

We construct a valid kernel deformed exponential family density from $Q$ and $f \in \mathcal{H}$. We first discuss the deformed log normalizer. In exponential family densities, the log-normalizer is the log of the normalizer. For deformed exponentials, the following holds.

**Lemma 1.** *Let $Z > 0$ be a constant. Then for $1 < \alpha \leq 2$,*

$$\frac{1}{Z}\exp_{2-\alpha}(Z^{\alpha-1}f(t)) = \exp_{2-\alpha}(f(t) - \log_\alpha Z)$$

*where*

$$\log_\beta t = \begin{cases} \frac{t^{1-\beta}-1}{1-\beta} \text{ if } t > 0, \beta \neq 1; \\ \log(t) \text{ if } t > 0, \beta = 1; \\ \text{undefined if } t \leq 0. \end{cases}$$

*Proof.* See Appendix B.1 $\qquad\qquad\square$

We describe a normalization sufficient condition analogous to Proposition 1 for the sparse deformed kernel exponential family. With Lemma 1, we can take an unnormalized $\exp_{2-\alpha}(\tilde{f}(t))$ and derive a valid normalized kernel deformed exponential family density. We only require that an affine function of the terms in the deformed-exponential are negative for large magnitude $t$.

**Proposition 2.** *For $1 < \alpha \leq 2$ assume $\tilde{p}(t) = \exp_{2-\alpha}(\tilde{f}(t))$ with $\tilde{f} \in \mathcal{H}$, $\mathcal{H}$ is a RKHS with kernel $k$. If $\exists C_t > 0$ s.t. for $\|t\|_2 > C_t$, $(\alpha-1)\tilde{f}(t) + 1 \leq 0$ and $k(t,t) \leq L_k\|t\|_2^\xi + C_k$ for some $\xi > 0$, then $\int_S \exp_{2-\alpha}(\tilde{f}(t))dQ < \infty$.*

*Proof.* See Appendix B.2 $\qquad\qquad\square$

We now construct a valid kernel deformed exponential family density using the finite integral.

**Corollary 2.** *Under the conditions of proposition 2, assume $\exp_{2-\alpha}(\tilde{f}(t)) > 0$ on a set $A \subseteq S$ such that $Q(A) > 0$, then $\exists$ constants $Z > 0$, $A_\alpha(f) \in \mathbb{R}$ such that for $f(t) = \frac{1}{Z^{\alpha-1}}\tilde{f}(t)$, the following holds*

$$\int_S \exp_{2-\alpha}(f(t) - A_\alpha(f))dQ = 1.$$

*Proof.* See Appendix B.3. $\qquad\qquad\square$

We thus estimate $\tilde{f}(t) = (Z)^{\alpha-1}f(t)$ and normalize to obtain a density of the desired form.

#### 4.2.2 APPROXIMATION THEORY

Under certain kernel conditions, kernel deformed exponential family densities can approximate densities of a similar form where the RKHS function is replaced with a $C_0(S)$ [1] function.

---

[1] continuous function on domain $S$ vanishing at infinity

**Proposition 3.** *Define*

$$\mathcal{P}_0 = \{\pi_f(t) = \exp_{2-\alpha}(f(t) - A_\alpha(f)), t \in S : f \in C_0(S)\}$$

*where $S \subseteq \mathbb{R}^d$. Suppose $k(x, \cdot) \in C_0(S), \forall x \in S$ and*

$$\int \int k(x, y)d\mu(x)d\mu(y) > 0, \forall \mu \in M_b(S)\backslash\{0\}. \tag{10}$$

*here $M_b(S)$ is the space of bounded measures over $S$. Then the set of deformed exponential families is dense in $\mathcal{P}_0$ wrt $L^r(Q)$ norm and Hellinger distance.*

*Proof.* See Appendix B.4 □

We apply this to approximate fairly general densities with kernel deformed exponential families.

**Theorem 1.** *Let $q_0 \in C(S)$, such that $q_0(t) > 0$ for all $t \in S$, where $S \subseteq \mathbb{R}^d$ is locally compact Hausdorff and $q_0(t)$ is the density of $Q$ with respect to a dominating measure $\nu$. Suppose there exists $l > 0$ such that for any $\epsilon > 0, \exists R > 0$ satisfying $|p(t) - l| \leq \epsilon$ for any $t$ with $\|t\|_2 > R$. Define*

$$\mathcal{P}_c = \{p \in C(S) : \int_S p(t)dQ = 1, p(t) \geq 0, \forall t \in S \text{ and } p - l \in C_0(S)\}.$$

*Suppose $k(t, \cdot) \in C_0(S)\forall t \in S$ and the kernel integration condition (Eqn. 10) holds. Then kernel deformed exponential families are dense in $\mathcal{P}_c$ wrt $L^r$ norm, Hellinger distance and Bregman divergence for the $\alpha$-Tsallis negative entropy functional.*

*Proof.* See Appendix B.5. □

For uniform $q_0$, kernel deformed exponential families can thus approximate continuous densities on compact domains arbitrarily well. Our Bregman divergence result is analagous to the KL divergence result in Sriperumbudur et al. (2017). KL divergence is Bregman divergence with the Shannon entropy functional: we show the same for Tsallis entropy. The Bregman divergence here describes how close the uncertainty in a density is to its first order approximation evaluated at another density. Using the Tsallis entropy functional here is appropriate for deformed exponential families: they maximize it given expected sufficient statistics (Naudts, 2004).

These results extend Sriperumbudur et al. (2017)'s approximation results to the deformed setting, where standard log and exponential rules cannot be applied. The Bregman divergence case requires bounding Frechet derivatives and applying the functional mean value theorem.

### 4.3 USING KERNELS FOR CONTINUOUS ATTENTION

We apply kernel exponential and deformed exponential families to attention. The forward pass computes attention densities and the context vector. The backwards pass uses automatic differentiation. We assume a vector representation $v \in \mathbb{R}^{|v|}$ computed from the locations we take an expectation over. For kernel exponential families we compute kernel weights $\{\gamma_i\}_{i=1}^I$ for $f(t) = \sum_{i=1}^I \gamma_i k(t, t_i)$

$$\gamma_i = w_i^T v,$$

and compute $Z = \int_S \exp(f(t))dQ$ numerically. For the deformed case we compute $\tilde{\gamma}_i = w_i^T v$ and $\tilde{f}(t) = (Z)^{\alpha-1} f(t) = \sum_{i=1}^I \tilde{\gamma}_i k(t, t_i)$ followed by $Z = \int_S \exp_{2-\alpha}(\tilde{f}(t))dQ$. The context

$$c = \mathbb{E}_{T \sim p}[V(T)] = \mathbf{B}\mathbb{E}_p[\Psi(t)]$$

requires taking the expectation of $\Psi(T)$ with respect to a (possibly deformed) kernel exponential family density $p$. Unlike Martins et al. (2020; 2021), where they obtained closed form expectations, difficult normalizing constants prevent us from doing so. We thus use numerical integration for the forward pass and automatic differentiation for the backward pass. Algorithm 1 shows how to compute a continuous attention mechanism for a kernel deformed exponential family attention density. The kernel exponential family case is similar.

---

**Algorithm 1** Continuous Attention Mechanism via Kernel Deformed Exponential Families

---

**Choose** base density $q_0(t)$ and kernel $k$. Inducing point locations $\{t_i\}_{i=1}^I$
**Input** Vector representation $v$ of input object i.e. document representation
**Parameters** $\{\tilde{\gamma}_i\}_{i=1}^I$ the weights for $\tilde{f}(t) = (Z)^{\alpha-1}f(t) = \sum_{i=1}^I \tilde{\gamma}_i k(t, t_i)$, matrix $\mathbf{B}$ for basis weights for value function $V(t) = \mathbf{B}\Psi(t)$. $I$ is number of inducing points.
**Forward Pass**
Compute $Z = \int \exp_{2-\alpha}(\tilde{f}(t))dQ(t)$ to obtain $p(t) = \frac{1}{Z}\exp_{2-\alpha}(\tilde{f}(t))$ via numerical integration
Compute $\mathbb{E}_{T\sim p}[\Psi(T)]$ via numerical integration
Compute $c = \mathbb{E}_{T\sim p}[V(T)] = \mathbf{B}\mathbb{E}_p[\Psi(T)]$
**Backwards Pass** use automatic differentiation

---

| Attention | N=32 | N=64 | N=128 | Mean |
|-----------|------|------|-------|------|
| Cts Softmax | 89.56 | 90.32 | **91.08** | 90.32 |
| Cts Sparsemax | 89.50 | 90.39 | 89.96 | 89.95 |
| Kernel Softmax | **91.30** | **91.08** | 90.44 | **90.94** |
| Kernel Sparsemax | 90.56 | 90.20 | 90.41 | 90.39 |

Table 1: IMDB sentiment classification dataset accuracy. Continuous softmax uses Gaussian attention, continuous sparsemax truncated parabola, and kernel softmax and sparsemax use kernel exponential and deformed exponential family with a Gaussian kernel. The latter has $\alpha = 2$ in exponential and multiplication terms. $N$: basis functions, $I = 10$ inducing points, bandwidth 0.01.

## 5 EXPERIMENTS

For document classification, we follow Martins et al. (2020)'s architecture. For the remaining, architectures have four parts: 1) an encoder takes a discrete representation of a time series and outputs attention density parameters. 2) The value function takes a time series representation (original or after passing through a neural network) and does (potentially multivariate) linear regression to obtain parameters $\mathbf{B}$ for a function $V(t; \mathbf{B})$. These are combined to compute 3) context $c = \mathbb{E}_p[V(T)]$, which is used in a 4) classifier. Fig. 2 in the Appendices visualizes this.

### 5.1 DOCUMENT CLASSIFICATION

We extend Martins et al. (2020)'s code[2] for the IMDB sentiment classification dataset (Maas et al., 2011). This starts with a document representation $v$ computed from a convolutional neural network and uses an LSTM attention model. We use a Gaussian base density and kernel, and divide the interval $[0, 1]$ into $I = 10$ inducing points where we evaluate the kernel in $f(t) = \sum_{i=1}^I \gamma_i k(t, t_i)$. We set the bandwidth to be 0.01 for $I = 10$. Table 1 shows results. On average, kernel exponential and deformed exponential family slightly outperforms the continuous softmax and sparsemax, although the results are essentially the same. The continuous softmax/sparsemax results are from their code.

---

[2]Martins et al. (2020)'s repository for this dataset is https://github.com/deep-spin/quati

---

| Attention | N=64 | N=128 | N=256 |
|-----------|------|-------|-------|
| Cts Softmax | 67.78±1.64 | 67.70± 2.49 | 68.00± 2.24 |
| Cts Sparsemax | 74.20±2.72 | 74.69±3.78 | 74.58±4.27 |
| Gaussian Mixture | 81.13±1.76 | 80.99±2.79 | 79.04±2.33 |
| Kernel Softmax | **93.85±0.60** | **94.26±0.75** | **93.83±0.60** |
| Kernel Sparsemax | 92.32±1.09 | 92.15±0.79 | 92.14±0.96 |

Table 2: Accuracy results on uWave gesture classification dataset for the irregularly sampled case. All methods use 100 attention heads. Gaussian mixture uses 100 components (and thus 300 parameters per head), and kernel methods use 256 inducing points.

| Attention | Accuracy | F1 |
|---|---|---|
| Cts Softmax | 96.97 | 83.69 |
| Cts Sparsemax | 96.04 | 73.71 |
| Gaussian Mixture | **97.20** | 84.89 |
| Kernel Softmax | 96.75 | 83.27 |
| Kernel Sparsemax | 96.86 | **84.90** |

Table 3: Accuracy results on MIT BIH Arrhythmia Classification dataset. For F1 score, kernel softmax and continuous softmax have similar results, while kernel sparsemax drastically outperforms continuous sparsemax. For accuracy, the Gaussian mixture slightly beats other methods.

## 5.2 uWave Dataset

We analyze uWave (Liu et al., 2009): accelerometer time series with eight gesture classes. We follow Li & Marlin (2016)'s split into 3,582 training observations and 896 test observations: sequences have length 945. We do synthetic irregular sampling and uniformly sample 10% of the observations. Table 2 shows results. Our highest accuracy is 94.26%, the unimodal case's best is 74.69%, and the mixture's best is 81.13%. Since this dataset is small, we report $\pm 1.96$ standard deviations from 10 runs. Fig. 1 shows that attention densities have support over non-overlapping time intervals. This cannot be done with Gaussian mixtures, and the intervals would be the same for each density in the exponential family case. Appendix C describes additional details.

## 6 ECG heartbeat classification

We use the MIT Arrhythmia Database's (Goldberger et al., 2000) Kaggle [3]. The task is to detect abnormal heart beats from ECG signals. The five different classes are {Normal, Supraventricular premature, Premature ventricular contraction, Fusion of ventricular and normal, Unclassifiable}. There are 87,553 training samples and 21,891 test samples. Our value function is trained using the output of repeated convolutional layers: the final layer has 256 dimensions and 23 time points. Our encoder is a feedforward neural network with the original data as input, and our classifier is a feedforward network. Table 3 shows results. All accuracies are very similar, but the $F1$ score of kernel sparsemax is drastically higher. Additional details are in Appendix D.

## 7 Discussion

In this paper we extend continuous attention mechanisms to use kernel exponential and deformed exponential family attention densities. The latter is a new flexible class of non-parametric densities with sparse support. We show novel existence properties for both kernel exponential and deformed exponential families, and prove approximation properties for the latter. We then apply these to the framework described in Martins et al. (2020; 2021) for continuous attention. We show results on three datasets: sentiment classification, gesture recognition, and arrhythmia classification. In the first case performance is similar to unimodal attention, for the second it is drastically better, and in the third it is similar in the dense case and drastically better in the sparse case.

### 7.1 Limitations and Future Work

A limitation of this work was the use of numerical integration, which scales poorly with the dimensionality of the locations. Because of this we restricted our applications to temporal and text data. This still allows for multiple observation dimensions at a given location. A future direction would be to use varianced reduced Monte Carlo to approximate the integral, as well as studying how to choose the number of basis functions in the value function and how to choose the number of inducing points.

---

[3]https://www.kaggle.com/mondejar/mitbih-database

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

## A  Proof Related to Proposition 1

### A.1  Proof of Proposition 1

*Proof.* This proof has several parts. We first bound the RKHS function $f$ and use the general tail bound we assumed to give a tail bound for the one dimensional marginals $T_{Qd}$ of $T_Q$. Using the RKHS function bound, we then bound the integral of the unnormalized density in terms of expectations with respect to these finite dimensional marginals. We then express these expectations over finite dimensional marginals as infinite series of integrals. For each integral within the infinite series, we use the finite dimensional marginal tail bound to bound it, and then use the ratio test to show that the infinite series converges. This gives us that the original unnormalized density has a finite integral.

We first note, following Wenliang et al. (2019), that the bound on the kernel in the assumption allows us to bound $f$ in terms of two constants and the absolute value of the point at which it is evaluated.

$$\begin{aligned}
|f(t)| &= |\langle f, k(t, \cdot)\rangle_{\mathcal{H}}| \text{ reproducing property} \\
&\leq \|f\|_{\mathcal{H}}\|k(t, \cdot)\|_{\mathcal{H}} \text{ Cauchy Schwarz} \\
&= \|f\|_{\mathcal{H}}\sqrt{\langle k(t, \cdot), k(t, \cdot)\rangle_{\mathcal{H}}} \\
&= \|f\|_{\mathcal{H}}\sqrt{k(t, t)} \\
&\leq \|f\|_{\mathcal{H}}\sqrt{L_k\|t\|^{\xi} + C_k} \text{ by assumption} \\
&\leq C_0 + C_1\|t\|^{|\xi|/2} \text{ for some } C_1, C_2 > 0.
\end{aligned}$$

We can write $T_Q = (T_{Q1}, \cdots, T_{QD})$. Let $e_d$ be a standard Euclidean basis vector. Then by the assumption and setting $u = e_d$ we have

$$P(|T_{Qd}| \geq z) \leq C_q \exp(-vz^{\eta})$$

Letting $Q_d$ be the marginal law,

$$\begin{aligned}
\int_S \exp(f(t))dQ &\leq \int_S \exp(C_0 + C_1\|t\|^{\xi/2})dQ \\
&= \exp(C_0)\int_S \exp(C_1\|t\|^{\xi/2})dQ \\
&= \exp(C_0)\mathbb{E}\exp(C_1\|T_Q\|^{\xi/2}) \\
&\leq \exp(C_0)\mathbb{E}\exp(C_1(\sqrt{d}\max_{d=1,\cdots,D}|T_{Qd}|)^{\xi/2}) \\
&\leq \exp(C_0)\sum_{d=1}^D \mathbb{E}\exp(C_2|T_{Qd}|^{\xi/2})
\end{aligned}$$

which will be finite if each $\mathbb{E}\exp(C_2|T_{Qd}|^{\xi/2}) < \infty$. Now letting $S_d$ be the relevant dimension of $S$,

$$\begin{aligned}
\mathbb{E}\exp(C_2|T_{Qd}|^{\xi/2}) &= \int_{S_d}\exp(C_2|t_d|^{\xi/2})dQ_d \\
&\leq \sum_{j=-\infty}^{-1}\int_j^{j+1}\exp(C_2|t_d|^{\xi/2})dQ_d + \sum_{j=0}^{\infty}\int_j^{j+1}\exp(C_2|t_d|^{\xi/2})dQ_d
\end{aligned}$$

where the inequality follows since $S_d \subseteq \mathbb{R}$, exp is a non-negative function and probability measures are monotonic. We will show that the second sum converges. Similar techniques can be shown for the first sum. Note that for $j \geq 0$

$$\begin{aligned}
Q_d([j, j+1)) &= P(T_d \geq j) - P(T_d \geq j+1) \\
&\leq P(T_d \geq j) \\
&\leq C_q\exp(-vj^{\eta}) \text{ by assumption}
\end{aligned}$$

Then

$$\sum_{j=0}^{\infty} \int_j^{j+1} \exp(C_2|t_d|^{\xi/2})dQ_d \le \sum_{j=0}^{\infty} \exp(C_2|j|^{\xi/2})Q_d([j,j+1))$$

$$\le \sum_{j=0}^{\infty} C_Q \exp(C_2|j|^{\xi/2} - vj^{\eta})$$

Let $a_j = \exp(C_2|j|^{\xi/2} - vi^{\eta})$. We will use the ratio test to show that the RHS converges. We have

$$\left|\frac{a_{j+1}}{a_j}\right| = \exp(C_2((j+1)^{\xi/2} - j^{\xi/2}) - v[(j+1)^{\eta} - j^{\eta}]). \tag{11}$$

We want this ratio to be $< 1$ for large $j$. We thus need to select $\eta$ so that for sufficiently large $j$, we have

$$\frac{C_1}{v}((j+1)^{\xi/2} - j^{\xi/2}) < [(j+1)^{\eta} - j^{\eta}].$$

Assume that $\eta > \frac{\xi}{2}$. Then

$$\frac{(j+1)^{\eta} - j^{\eta}}{(j+1)^{\xi/2} - j^{\xi/2}} = \frac{j^{\eta}[(1+\frac{1}{j})^{\eta} - 1]}{j^{\xi/2}[(1+\frac{1}{j})^{\xi/2} - 1]}$$

$$\ge j^{\eta - \xi/2}.$$

Since the RHS is unbounded for $\eta > \frac{\xi}{2}$, we have that Eqn. 11 holds for sufficiently large $j$. By the ratio test $\mathbb{E}_{q_d(t)} \exp(C_2|T_d|^{\xi/2}) = \sum_{j=-\infty}^{-1} \int_j^{j+1} \exp(C_2|t_d|^{\xi/2})dQ_d + \sum_{j=0}^{\infty} \int_j^{j+1} \exp(C_2|t_d|^{\xi/2})dQ_d$ is finite. Thus putting everything together we have

$$\int_S \exp(f(t))dQ \le \int_S \exp(C_0 + C_1\|t\|^{\xi/2})dQ$$

$$< \exp(C_0)\sum_{d=1}^{D} \mathbb{E} \exp(C_2|T_{Qd}|^{\xi/2})$$

$$< \infty$$

and $\tilde{p}(t)$ can be normalized.

$\square$

## A.2 PROOF OF COROLLARY 1

*Proof.* Let $\xi = 4$. Then $\eta > 2$ and

$$P(|u^T T| > t) \le P(|u^T T| \ge t) \text{ monotonicity}$$

$$\le C_Q \exp(-vt^{\eta})$$

$$< C_Q \exp(-vt^2).$$

The second case is similar. For the uniformly bounded kernel,

$$\int_S \exp(\langle f, k(\cdot, t)\rangle_{\mathcal{H}})dQ \le \exp(\|f\|_{\mathcal{H}}\sqrt{C_k})\int_S dQ$$

$$= \exp(\|f\|_{\mathcal{H}}\sqrt{C_k})$$

$$< \infty.$$

The first line follows from Cauchy Schwarz and $\xi = 0$

$\square$

# B    PROOFS RELATED TO KERNEL DEFORMED EXPONENTIAL FAMILY

## B.1    PROOF OF LEMMA 1

*Proof.* The high level idea is to express a term inside the deformed exponential family that becomes $1/Z$ once outside.

$$
\begin{aligned}
\exp_{2-\alpha}(f(t) - \log_\alpha(Z)) &= [1 + (\alpha - 1)(f(t) - \log_\alpha Z)]_+^{\frac{1}{\alpha-1}} \\
&= [1 + (\alpha - 1)f(t) - (\alpha - 1)\frac{Z^{1-\alpha} - 1}{1 - \alpha}]_+^{\frac{1}{\alpha-1}} \\
&= [1 + (\alpha - 1)f(t) + Z^{1-\alpha} - 1]_+^{\frac{1}{\alpha-1}} \\
&= [(\alpha - 1)f(t) + Z^{1-\alpha}]_+^{\frac{1}{\alpha-1}} \\
&= [(\alpha - 1)f(t)\frac{Z^{\alpha-1}}{Z^{\alpha-1}} + Z^{1-\alpha})]_+^{\frac{1}{\alpha-1}} \\
&= \frac{1}{Z}[(\alpha - 1)f(t)\frac{1}{Z^{\alpha-1}} + 1]_+^{\frac{1}{\alpha-1}} \\
&= \frac{1}{Z}\exp_{2-\alpha}(Z^{\alpha-1}f(t))
\end{aligned}
$$

$\square$

## B.2    PROOF OF PROPOSITION 2

*Proof.*

$$
\begin{aligned}
\int_S \exp_{2-\alpha}(\tilde{f}(t))dQ &= \int_S [1 + (\alpha - 1)\tilde{f}(t)]_+^{\frac{1}{\alpha-1}} dQ \\
&= \int_{\|t\| \leq C_t} [1 + (\alpha - 1)\tilde{f}(t)]_+^{\frac{1}{\alpha-1}} dQ \\
&\leq \int_{\|t\| \leq C_t} [1 + (\alpha - 1)(C_0 + C_1|C_t|^{\xi/2})]_+^{\frac{1}{\alpha-1}} dQ \\
&< \infty
\end{aligned}
$$

$\square$

## B.3    PROOF OF COROLLARY 2

*Proof.* From proposition 2 and the assumption,

$$
\int_S \exp_{2-\alpha}(\tilde{f}(t))dQ = Z
$$

for some $Z > 0$. Then

$$
\int_S \frac{1}{Z}\exp_{2-\alpha}(Z^{\alpha-1}f(t))dQ = 1
$$

$$
\int_S \exp_{2-\alpha}(f(t) - \log_\alpha Z)dQ = 1
$$

where the second line follows from lemma 1. Set $A_\alpha(f) = \log_\alpha(Z)$ and we are done.    $\square$

## B.4    PROOF OF PROPOSITION 3

*Proof.* The kernel integration condition tells us that $\mathcal{H}$ is dense in $C_0(S)$ with respect to $L^\infty$ norm. This was shown in Sriperumbudur et al. (2011). For the $L^r$ norm, we apply $\|p_f - p_g\|_{L^r} \leq 2M_{\exp}\|f - g\|_\infty$ from Lemma 5 with $f \in C_0(S)$, $g \in \mathcal{H}$, and $f_0 = f$. $L^1$ convergence implies Hellinger convergence.    $\square$

## B.5 Proof of Theorem 1

*Proof.* For any $p \in \mathcal{P}_c$, define $p_\delta = \frac{p+\delta}{1+\delta}$. Then

$$\|p - p_\delta\|_r = \frac{\delta}{1+\delta}\|p\|_r$$
$$\to 0$$

for $1 \le r \le \infty$. Thus for any $\epsilon > 0, \exists \delta_\epsilon > 0$ such that for any $0 < \theta < \delta_\epsilon$, we have $\|p - p_\theta\|_r \le \epsilon$, where $p_\theta(t) > 0$ for all $t \in S$.

Define $f = \left(\frac{1+\theta}{l+\theta}\right)^{1-\alpha} \log_{2-\alpha} p_\theta \frac{1+\theta}{l+\theta}$. Since $p \in C(S)$, so is $f$. Fix any $\eta > 0$ and note that

$$f(t) \ge \eta$$
$$\left(\frac{1+\theta}{l+\theta}\right)^{1-\alpha} \log_{2-\alpha} p_\theta \frac{1+\theta}{l+\theta} \ge \eta$$
$$\log_{2-\alpha} p_\theta \frac{1+\theta}{l+\theta} \ge \left(\frac{1+\theta}{l+\theta}\right)^{\alpha-1} \eta$$
$$p_\theta \frac{1+\theta}{l+\theta} \ge \exp_{2-\alpha}\left(\left(\frac{1+\theta}{l+\theta}\right)^{\alpha-1} \eta\right)$$
$$p_\theta \ge \frac{l+\theta}{1+\theta} \exp_{2-\alpha}\left(\left(\frac{1+\theta}{l+\theta}\right)^{\alpha-1} \eta\right)$$
$$p - l \ge (l+\theta)\left(\exp_{2-\alpha}\left(\left(\frac{1+\theta}{l+\theta}\right)^{\alpha-1} \eta\right) - 1\right)$$

Thus

$$A = \{t : f(t) \ge \eta\}$$
$$= \left\{p - l \ge (l+\theta)\left(\exp_{2-\alpha}\left(\left(\frac{1+\theta}{l+\theta}\right)^{\alpha-1} \eta\right) - 1\right)\right\}$$

Since $p - l \in C_0(S)$ the set on the second line is bounded. Thus $A$ is bounded so that $f \in C_0(S)$. Further, by Lemma 1

$$p_\theta = \exp_{2-\alpha}\left(f - \log_\alpha \frac{1+\theta}{l+\theta}\right)$$

giving us $p_\theta \in \mathcal{P}_0$. By Proposition 3 there is some $p_g$ in the deformed kernel exponential family so that $\|p_\theta - p_g\|_{L^r(S)} \le \epsilon$. Thus $\|p - p_g\|_r \le 2\epsilon$ for any $1 \le r \le \infty$. To show convergence in Helinger distance, note

$$H^2(p, p_g) = \frac{1}{2}\int_S (\sqrt{p} - \sqrt{p_g})^2 dQ$$
$$= \frac{1}{2}\int_S (p - 2\sqrt{pp_g} + p_g) dQ$$
$$\le \frac{1}{2}\int_S (p - 2\min(p, p_g) + p_g) dQ$$
$$= \frac{1}{2}\int_S |p - p_g| dQ$$
$$= \frac{1}{2}\|p - p_g\|_1$$

---

[4] actually an equality, see https://www2.cs.uic.edu/ zhangx/teaching/bregman.pdf for proof

so that $L^1(S)$ convergence, which we showed, implies Hellinger convergence. Let us consider the Bregman divergence. Note the generalized triangle inequality[4] for Bregman divergence

$$B_{\Omega_\alpha}(p, p_g) = \underbrace{B_{\Omega_\alpha}(p, p_\theta)}_{I} + \underbrace{B_{\Omega_\alpha}(p_\theta, p_g)}_{II} - \underbrace{\langle p - p_\theta, \nabla\Omega_\alpha(p_\theta) - \nabla\Omega_\alpha(p_g)\rangle_2}_{III} \qquad (12)$$

**Term I**

$$\begin{aligned}
B_{\Omega_\alpha}(p, p_\theta) &= \frac{1}{\alpha(\alpha-1)} \int_S (p^\alpha - p_\theta^\alpha)dQ - \langle \nabla\Omega_\alpha(p_\theta), p - p_\theta\rangle \\
&= \frac{1}{\alpha(\alpha-1)} \int_S (p^\alpha - p_\theta^\alpha)dQ - \frac{1}{\alpha-1} \int p_\theta^{\alpha-1}(p - p_\theta)dQ \\
&\leq \frac{1}{\alpha(\alpha-1)} \int_S (p^\alpha - p_\theta^\alpha)dQ + \frac{1}{\alpha-1}\|p_\theta^{\alpha-1}\|_1\|\|p - p_\theta\|_\infty
\end{aligned}$$

The first term on the rhs clearly vanishes as $\theta \to 0$. For the second term, we already showed that $\|p - p_\theta\|_\infty \to 0$.

**Term II**

Fix $\theta$. Then term $II$ converges to 0 by Lemma 5.

**Term III**

For term $III$,

$$\langle p - p_\theta, \nabla\Omega_\alpha(p_\theta) - \nabla\Omega_\alpha(p_g)\rangle_2 \leq \|p - p_\theta\|_\infty\|\nabla\Omega_\alpha(p_\theta) - \nabla\Omega_\alpha(p_g)\|_1$$

Clearly the first term on the rhs converges by $L^r$ convergence. The $L^1$ term for the gradient is given by

$$\begin{aligned}
\|\nabla\Omega_\alpha(p_\theta) - \nabla\Omega_\alpha(p_g)\|_1 &= \frac{1}{\alpha-1} \int |p_\theta(t)^{\alpha-1} - p_g(t)^{\alpha-1}|dQ \\
&\leq \int (\|p_\theta\|_\infty + \|p_\theta - p_g\|_\infty)^{\alpha-2}\|p_\theta - p_g\|_\infty dQ \quad \text{Eqn. 17} \\
&= (\|p_\theta\|_\infty + \|p_\theta - p_g\|_\infty)^{\alpha-2}\|p_\theta - p_g\|_\infty
\end{aligned}$$

so that the inner product terms are bounded as

$$|\langle p - p_\theta, \nabla\Omega_\alpha(p_\theta) - \nabla\Omega_\alpha(p_g)\rangle_2| \leq (\|p_\theta\|_\infty + \|p_\theta - p_g\|_\infty)^{\alpha-2}\|p_\theta - p_g\|_\infty\|p - p_\theta\|_\infty$$

$\square$

**Lemma 2.** *(Functional Taylor's Theorem) Let $F : X \to \mathbb{R}$ where $X$ is a Banach space. Let $f, g \in X$ and let $F$ be $k$ times Gateaux differentiable. Then we can write*

$$F(g) = \sum_{i=0}^{k-1} F^i(f)(g - f)^i + F^k(f + c(g - f))(g - f)^k$$

*for some $c \in [0, 1]$.*

*Proof.* This is simply a consequence of expressing a functional as a function of an $\epsilon \in [0, 1]$, which restricts the functional to take input functions between two functions. We then apply Taylor's theorem to the function and apply the chain rule for Gateaux derivatives to obtain the resulting Taylor remainder theorem for functionals.

Let $G(\eta) = F(f + \eta(g - f))$. By Taylor's theorem we have

$$G(1) = G(0) + G'(0) + \cdots + G^k(c)$$

and applying the chain rule gives us

$$F(g) = \sum_{i=0}^{k-1} F^i(f)(g - f)^i + F^k(f + c(g - f))(g - f)^k$$

$\square$

**Lemma 3.** *(Functional Mean Value Theorem) Let $F : X \to \mathbb{R}$ be a functional where $f, g \in X$ some Banach space with norm $\| \cdot \|$. Then*

$$|F(f) - F(g)| \leq \|F'(h)\|_{op}\|f - g\|$$

*where $h = g + c(f - g)$ for some $c \in [0, 1]$, $F'(h)$ is the Gateaux derivative of $F$, and $\| \cdot \|_{op}$ is the operator norm $\|A\|_{op} = \inf\{c > 0 : \|Ax\| \leq c\|x\| \forall x \in X\}$.*

*Proof.* Consider $G(\eta) = F(g + \eta(f - g))$. Apply the ordinary mean value theorem to obtain

$$G(1) - G(0) = G'(c), c \in [0, 1]$$
$$= F'(g + c(f - g)) \cdot (f - g)$$

and thus

$$|F(f) - F(g)| \leq \|F'(h)\|_{op}\|f - g\|$$

$\square$

**Claim 1.** *Consider $\mathcal{P}_\infty = \{p_f = \exp_{2-\alpha}(f - A_\alpha(f)) : f \in L^\infty(S)\}$. Then for $p_f \in \mathcal{P}_\infty$, $A_\alpha(f) \leq \|f\|_\infty$.*

*Proof.*

$$p_f(t) = \exp_{2-\alpha}(f(t) - A_\alpha(f))$$
$$\leq \exp_{2-\alpha}(\|f\|_\infty - A_\alpha(f)) \text{ for } 1 < \alpha \leq 2$$
$$\int_S p_f(t)dQ \leq \int_S \exp_{2-\alpha}(\|f\|_\infty - A_\alpha(f))dQ$$
$$1 \leq \exp_{2-\alpha}(\|f\|_\infty - A_\alpha(f))$$
$$\log_{2-\alpha} 1 \leq \|f\|_\infty - A_\alpha(f)$$
$$A_\alpha(f) \leq \|f\|_\infty$$

where for the second line recall that we assumed that throughout the paper $1 < \alpha \leq 2$. $\square$

**Lemma 4.** *Consider $\mathcal{P}_\infty = \{p_f = \exp_{2-\alpha}(f - A_\alpha(f)) : f \in L^\infty(S)\}$. Then the Frechet derivative of $A_\alpha : L^\infty \to \mathbb{R}$ exists. It is given by the map*

$$A'(f)(g) = \mathbb{E}_{\tilde{p}_f^{2-\alpha}}(g(T))$$
$$= \frac{\int p_f^{2-\alpha}(t)g(t)dQ}{\int p_f^{2-\alpha}(t)dQ}$$

*Proof.* This proof has several parts. We first derive the Gateaux differential of $p_f$ in a direction $\psi \in L^\infty$ and as it depends on the Gateaux differential of $A_\alpha(f)$ in that direction, we can rearrange terms to recover the latter. We then show that it exists for any $f, \psi \in L^\infty$. Next we show that the second Gateaux differential of $A_\alpha(f)$ exists, and use that along with a functional Taylor expansion to prove that the first Gateaux derivative is in fact a Frechet derivative.

In Martins et al. (2020) they show how to compute the gradient of $A_\alpha(\theta)$ for the finite dimensional case: we extend this to the Gateaux differential. We start by computing the Gateaux differential of $p_f$.

$$\frac{d}{d\eta}p_{f+\eta\psi}(t) = \frac{d}{d\eta}\exp_{2-\alpha}(f(t) + \eta\psi(t) - A_\alpha(f + \eta\psi))$$

$$= \frac{d}{d\eta}[1 + (\alpha - 1)(f(t) + \eta\psi(t) - A_\alpha(f + \eta\psi))]_+^{1/(\alpha-1)}$$

$$= [1 + (\alpha - 1)(f(t) + \eta\psi(t) - A_\alpha(f + \eta\psi))]_+^{(2-\alpha)/(\alpha-1)}\left(\psi(t) - \frac{d}{d\eta}A_\alpha(f + \eta\psi)\right)$$

$$= p_{f+\eta\psi}^{2-\alpha}(t)\left(\psi(t) - \frac{d}{d\eta}A_\alpha(f + \eta\psi)\right)$$

evaluating at $\eta = 0$ gives us

$$dp(f; \psi)(t) = p_f^{2-\alpha} \left( \psi(t) + dA_\alpha(f; \psi) \right)$$

Note that by claim 1 we have

$$\begin{aligned}
p_{f+\eta\psi}(t) &= \exp_{2-\alpha}(f(t) + \eta\psi(t) - A_\alpha(f + \eta\psi(t))) \\
&\leq \exp_{2-\alpha}(2\|f\|_\infty + 2\eta\|\psi\|_\infty) \\
&\leq \exp_{2-\alpha}(2(\|f\|_\infty + \|\psi\|_\infty))
\end{aligned}$$

We can thus apply the dominated convergence theorem to pull a derivative with respect to $\eta$ under an integral. We can then recover the Gateaux diferential of $A_\alpha$ via

$$\begin{aligned}
0 &= \frac{d}{d\eta}\bigg|_{\eta=0} \int p_{f+\eta\psi}(t)dQ \\
&= \int dp(f; \psi)(t)dQ \\
&= \int p_f(t)^{2-\alpha}(\psi(t) - dA_\alpha(f; \psi))dQ \\
dA_\alpha(f; \psi) &= \mathbb{E}_{\tilde{p}_f^{2-\alpha}}(\psi(T)) \\
&< \infty
\end{aligned}$$

where the last line follows as $\psi \in L^\infty$. Thus the Gateaux derivative exists in $L^\infty$ directions. The derivative at $f$ maps $\psi :\rightarrow \mathbb{E}_{\tilde{p}_f^{2-\alpha}}(\psi(T))$ i.e. $A'_\alpha(f)(\psi) = \mathbb{E}_{\tilde{p}_f^{2-\alpha}}(\psi(T))$. We need to show that this is a Frechet derivative. To do so, we will take take second derivatives of $p_{f+\eta\psi}(t)$ with respect to $\eta$ in order to obtain second derivatives of $A_\alpha(f + \eta\psi)$ with respect to $\eta$. We will then construct a functional second order Taylor expansion. By showing that the second order terms converge sufficiently quickly, we will prove that the map $\psi :\rightarrow \mathbb{E}_{\tilde{p}_f^{2-\alpha}}(\psi(T))$ is a Frechet derivative.

$$\begin{aligned}
\frac{d^2}{d\eta^2}p_{f+\eta\psi}(t) &= \frac{d}{d\eta}p_{f+\eta\psi}(t)^{2-\alpha}\left(\psi(t) - \frac{d}{d\eta}A_\alpha(f + \eta\psi)\right) \\
&= \left(\frac{d}{d\eta}p_{f+\eta\psi}(t)^{2-\alpha}\right)\left(\psi(t) - \frac{d}{d\eta}A_\alpha(f + \eta\psi)\right) \\
&\quad - p_{f+\eta\psi}(t)^{2-\alpha}\frac{d^2}{d\eta^2}A_\alpha(f + \eta\psi) \\
&= (2-\alpha)p_{f+\eta\psi}(t)(\psi(t) - \frac{d}{d\eta}A_\alpha(f + \eta\psi))\frac{d}{d\eta}p_{f+\eta\psi}(t) \\
&\quad - p_{f+\eta\psi}(t)^{2-\alpha}\frac{d^2}{d\eta^2}A_\alpha(f + \eta\psi) \\
&= (2-\alpha)p_{f+\eta\psi}^{3-2\alpha}(\psi(t) - \frac{d}{d\eta}A_\alpha(f + \eta\psi))^2 - p_{f+\eta\psi}(t)^{2-\alpha}\frac{d^2}{d\eta^2}A_\alpha(f + \eta\psi)
\end{aligned}$$

We need to show that we can again pull the second derivative under the integral. We already showed that we can pull the derivative under once (for the first derivative) and we now need to do it again. We need to show an integrable function that dominates $p_{f+\eta\psi}(t)^{2-\alpha}(\psi(t) - \mathbb{E}_{\tilde{p}_{f+\eta\psi}^{2-\alpha}}\psi(T))$.

$$\begin{aligned}
|p_{f+\eta\psi}(t)^{2-\alpha}(\psi(t) - \mathbb{E}_{\tilde{p}_{f+\eta\psi}^{2-\alpha}}\psi(T))| &\leq p_{f+\eta\psi}(t)^{2-\alpha}2\|\psi\|_\infty \\
&\leq \exp_{2-\alpha}(2(\|f\|_\infty + \|\psi\|_\infty))2\|\psi\|_\infty
\end{aligned}$$

which is in $L^1(Q)$. Now applying the dominated convergence theorem

$$\begin{aligned}
0 &= \int \frac{d^2}{d\epsilon^2}p_{f+\epsilon\psi}(t)dQ \\
&= \int \left[ (2-\alpha)p_{f+\epsilon\psi}^{3-2\alpha}(\psi(t) - \frac{d}{d\epsilon}A_\alpha(f + \epsilon\psi))^2 - p_{f+\epsilon\psi}(t)^{2-\alpha}\frac{d^2}{d\epsilon^2}A_\alpha(f + \epsilon\psi) \right] dQ
\end{aligned}$$

and rearranging gives

$$\frac{d^2}{d\epsilon^2} A_\alpha(f + \epsilon\psi) = (2 - \alpha) \frac{\int p_{f+\epsilon\psi}^{3-2\alpha}(\psi(t) - \frac{d}{d\epsilon}A_\alpha(f + \epsilon\psi))^2 dQ}{\int p_{f+\epsilon\psi}(t)^{2-\alpha} dQ}$$

$$\left.\frac{d^2}{d\epsilon^2} A_\alpha(f)\right|_{\epsilon=0} = (2 - \alpha) \frac{\int p_f^{3-2\alpha}(\psi(t) - \mathbb{E}_{\tilde{p}_f^{2-\alpha}}[\psi(T)])^2 dQ}{\int p_f(t)^{2-\alpha} dQ}$$

since $f, \psi \in L^\infty$. For the functional Taylor expansion, we have from Lemma 2

$$A_\alpha(f + \psi) = A_\alpha(f) + A'_\alpha(f)(\psi) + \frac{1}{2}A''_\alpha(f + \epsilon\psi)(\psi)^2$$

for some $\epsilon \in [0,1]$. We thus need to show that for $\epsilon \in [0,1]$,

$$(2 - \alpha) \frac{\frac{1}{\|\psi\|_\infty}\int p_{f+\epsilon\psi}^{3-2\alpha}(\psi(t) - \mathbb{E}_{\tilde{p}_{f+\epsilon\psi}^{2-\alpha}}[\psi(T)])^2 dQ}{\int p_{f+\epsilon\psi}(t)^{2-\alpha} dQ} \overset{\psi \to 0}{\to} 0$$

It suffices to show that the numerator tends to 0 as $\psi \to 0$.

$$\left| \frac{1}{\|\psi\|_\infty}(\psi(t) - \mathbb{E}_{\tilde{p}_{f+\epsilon\psi}^{2-\alpha}}[\psi(T)])^2 \right|$$

$$= \left| \frac{\psi(t)}{\|\psi\|_\infty}\psi(t) - \frac{\psi(t)}{\|\psi\|_\infty}2\mathbb{E}_{\tilde{p}_{f+\epsilon\psi}^{2-\alpha}}[\psi(T)] + \frac{\mathbb{E}_{\tilde{p}_{f+\epsilon\psi}^{2-\alpha}}[\psi(T)]}{\|\psi\|_\infty}\mathbb{E}_{\tilde{p}_{f+\epsilon\psi}^{2-\alpha}}[\psi(T)] \right|$$

$$\leq \left| \frac{\psi(t)}{\|\psi\|_\infty} \right| \left| \psi(t) - 2\mathbb{E}_{\tilde{p}_{f+\epsilon\psi}^{2-\alpha}}[\psi(T)] \right|$$

$$+ \left| \mathbb{E}_{\tilde{p}_{f+\epsilon\psi}^{2-\alpha}}\frac{\psi(T)}{\|\psi\|_\infty} \right| \left| \mathbb{E}_{\tilde{p}_{f+\epsilon\psi}^{2-\alpha}}[\psi(T)] \right|$$

$$\leq \left| \psi(t) - 2\mathbb{E}_{\tilde{p}_{f+\epsilon\psi}^{2-\alpha}}[\psi(T)] \right| + \|p_{f+\epsilon\psi}\|_{2-\alpha}^{2-\alpha} \left| \mathbb{E}_{\tilde{p}_{f+\epsilon\psi}^{2-\alpha}}[\psi(T)] \right|$$

$$\to 0 \text{ as } \psi \to 0$$

and plugging this in we obtain the desired result. Thus the Frechet derivative of $A_\alpha(f)$ exists. $\qquad\square$

**Lemma 5.** *Define $\mathcal{P}_\infty = \{p_f = \exp_{2-\alpha}(f - A_\alpha(f)) : f \in L^\infty(S)\}$ where $L^\infty(S)$ is the space of almost surely bounded measurable functions with domain $S$. Fix $f_0 \in L^\infty$. Then for any fixed $\epsilon > 0$ and $p_g, p_f \in \mathcal{P}_\infty$ such that $f, g \in \overline{B}_\epsilon^\infty(f_0)$ the $L^\infty$ closed ball around $f_0$, there exists constant $M_{\exp} > 0$ depending only on $f_0$ such that*

$$\|p_f - p_g\|_{L^r} \leq 2M_{\exp}\|f - g\|_\infty$$

*Further*

$$B_{\Omega_\alpha}(p_f, p_g) \leq \frac{1}{\alpha - 1}\|p_f - p_g\|_\infty[(\|p_f\|_\infty + \|p_f - p_g\|_\infty)^{\alpha-1} + \exp_{2-\alpha}(2\|g\|_\infty)]$$

*Proof.* This Lemma mirrors Lemma A.1 in Sriperumbudur et al. (2017), but the proof is very different as they rely on the property that $\exp(x + y) = \exp(x)\exp(y)$, which does not hold for $\beta$-exponentials. We thus had to strengthen the assumption to include that $f$ and $g$ lie in a closed ball, and then use the functional mean value theorem Lemma 3 as the main technique to achieve our result.

Consider that by functional mean value inequality,

$$\|p_f - p_g\|_{L^r} = \|\exp_\beta(f - A_\alpha(f)) - \exp_\beta(g - A_\alpha(g))\|_{L^r}$$

$$\leq \|\exp_\beta(h - A_\alpha(h))^{2-\alpha}\|_\infty(\|f - g\|_\infty + |A_\alpha(f) - A_\alpha(g)|) \quad (13)$$

where $h = cf + (1 - c)g$ for some $c \in [0,1]$. We need to bound $\exp_\beta(h - A_\alpha(h))$ and $\|A_\alpha(f) - A_\alpha(g)\|_\infty$.

We can show a bound on $\|h\|_\infty$

$$
\begin{aligned}
\|h\|_\infty &= \|cf + (1-c)g - f_0 + f_0\|_\infty \\
&\leq \|c(f - f_0) + (1-c)(g - f_0) + f_0\|_\infty \\
&\leq c\|f - f_0\|_\infty + (1-c)\|g - f_0\|_\infty + \|f_0\|_\infty \\
&\leq \epsilon + \|f_0\|_\infty
\end{aligned}
$$

so that $h$ is bounded. Now we previously showed in claim 1 that $|A_\alpha(h)| \leq \|h\|_\infty \leq \epsilon + \|f_0\|_\infty$. Since $h, A_\alpha(h)$ are both bounded $\exp_\beta(h - A_\alpha(h))^{2-\alpha}$ is also.

Now note that by Lemma 3,

$$
|A_\alpha(f) - A_\alpha(g)| \leq \|A'_\alpha(h)\|_{\text{op}}\|f - g\|_\infty
$$

We need to show that $\|A'_\alpha(h)\|_{\text{op}}$ is bounded for $f, g \in \overline{B}_\epsilon(f_0)$. Note that in Lemma 4 we showed that

$$
|A'_\alpha(f)(g)| = |\mathbb{E}_{p_f^{2-\alpha}}[g(T)]|
$$

$$
\leq \|g\|_\infty
$$

Thus $\|A'_\alpha\|_{\text{op}} = \sup\{|A'_\alpha(h)(m)| : \|m\|_\infty = 1\} \leq 1$. Let $M_{\exp}$ be the bound on $\exp_\beta(h - A_\alpha(h))$. Then putting everything together we have the desired result

$$
\|p_f - p_g\|_{L^r} \leq 2M_{\exp}\|f - g\|_\infty
$$

Now

$$
B_{\Omega_\alpha}(p_f, p_g) = \Omega_\alpha(p_f) - \Omega_\alpha(p_g) - \langle \nabla\Omega_\alpha(p_g), p_f - p_g \rangle_2 \tag{14}
$$

For the inner prodct term, first note that following Martins et al. (2020) the gradient is given by

$$
\nabla\Omega_\alpha(p_g)(t) = \frac{p_g(t)^{\alpha-1}}{\alpha - 1} \tag{15}
$$

Thus

$$
\begin{aligned}
|\langle \nabla\Omega_\alpha(p_g), p_f - p_g \rangle_2| &\leq \|\nabla\Omega_\alpha(p_g)\|_1\|p_f - p_g\|_\infty \\
&= \frac{1}{\alpha - 1}\int_S \exp_{2-\alpha}(g(t) - A(g))dQ\|p_f - p_g\|_\infty \\
&\leq \frac{1}{\alpha - 1}\exp_{2-\alpha}(2\|g\|_\infty)\|p_f - p_g\|_\infty
\end{aligned}
$$

where the second line follows from claim 1.

Further note that by Taylor's theorem,

$$
y^\alpha = x^\alpha + \alpha z^{\alpha-1}(y - x) \tag{16}
$$

for some $z$ between $x$ and $y$. Then letting $y = p_f(t)$ and $x = p_g(t)$, we have for some $z = h(t)$ lying between $p_f(t)$ and $p_g(t)$ that

$$
p_f(t)^\alpha = p_g(t)^\alpha + \alpha h(t)^{\alpha-1}(p_f(t) - p_g(t))
$$

Since $f \in L^\infty$ then applying Claim 1 we have that each $p_f, p_g \in L^\infty$ and thus $h$ is. Then

$$
\begin{aligned}
|p_f(t)^\alpha - p_g(t)^\alpha| &= \alpha|h(t)|^{\alpha-1}|p_f(t) - p_g(t)| \\
&\leq \alpha\|h\|_\infty^{\alpha-1}\|p_f - p_g\|_\infty \\
&\leq \alpha\max\{\|p_f\|_\infty, \|p_g\|_\infty\}^{\alpha-1}\|p_f - p_g\|_\infty \\
&\leq \alpha(\|p_f\|_\infty + \|p_f - p_g\|_\infty)^{\alpha-1}\|p_f - p_g\|_\infty \tag{17}
\end{aligned}
$$

so that

$$
\begin{aligned}
|\Omega_\alpha(p_f) - \Omega_\alpha(p_g)| &= \left|\frac{1}{\alpha(\alpha-1)}\int(p_f(t)^\alpha - p_g(t)^\alpha)dQ\right| \\
&\leq \frac{1}{\alpha - 1}(\|p_f\|_\infty + \|p_f - p_g\|_\infty)^{\alpha-1}\|p_f - p_g\|_\infty.
\end{aligned}
$$

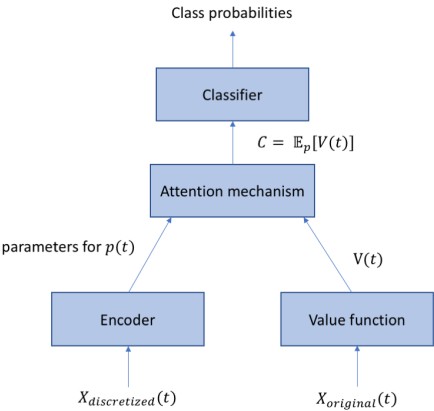

Figure 2: General architecture for classification using continuous attention mechanisms. The pipeline is trained end-to-end. The encoder takes a discretized representation of an observation (i.e. a time series) and outputs parameters for an attention density. The value function takes the original (potentially irregularly sampled) time series and outputs parameters for a function $V(t)$. These are then combined in an attention mechanism by computing a context vector $c = \mathbb{E}_p[V(T)]$. For some parametrizations of $p$ and $V(t)$ this can be computed in closed form, while for others it must be done via numerical integration. The context vector is then fed into a classifier.

Putting it all together we obtain

$$
\begin{aligned}
B_{\Omega_\alpha}(p_f, p_g) &\leq \frac{1}{\alpha - 1}(\|p_f\|_\infty + \|p_f - p_g\|_\infty)^{\alpha-1}\|p_f - p_g\|_\infty \\
&+ \frac{1}{\alpha - 1}\exp_{2-\alpha}(2\|g\|_\infty)\|p_f - p_g\|_\infty \\
&= \frac{1}{\alpha - 1}\|p_f - p_g\|_\infty[(\|p_f\|_\infty + \|p_f - p_g\|_\infty)^{\alpha-1} + \exp_{2-\alpha}(2\|g\|_\infty)]
\end{aligned}
$$

$\square$

## C  uWave Experiments: Additional Details

We experiment with $N = 64, 128$ and $256$ basis functions, and use a learning rate of $1e - 4$. We use $H = 100$ attention mechanisms, or heads. Unlike Vaswani et al. (2017), our use of multiple heads is slightly different as we use the same value function for each head, and only vary the attention densities. Additional architectural details are given below.

### C.1  Value Function

The value function uses regularized linear regression on the original time series observed at random observation times (which are not dependent on the data) to obtain an approximation $V(t; \mathbf{B}) = \mathbf{B}\Psi(t) \approx X(t)$. The $H$ in Eqn. 6 is the original time series.

#### C.1.1  Encoder

In the encoder, we use the value function to interpolate the irregularly sampled time series at the original points. This is then passed through a convolutional layer with 4 filters and filter size 5 followed by a max pooling layer with pool size 2. This is followed by one hidden layer with 256

units and an output $v$ of size 256. The attention densities for each head $h = 1, \cdots, H$ are then

$$\mu_h = w_{h,1}^T v$$
$$\sigma_h = \text{softplus}(w_{h,2}^T v)$$
$$\gamma_h = W^{(h)} v$$

for vectors $w_{h,1}, w_{h,2}$ and matrices $W^h$ and heads $h = 1, \cdots, H$

### C.1.2 ATTENTION MECHANISM

After forming densities and normalizing, we have densities $p_1(t), \cdots, p_H(t)$, which we use to compute context scalars

$$c_h = \mathbb{E}_{p_h}[V(T)]$$

We compute these expectations using numerical integration to compute basis function expectations $\mathbb{E}_{p_h}[\psi_n(T)]$ and a parametrized value function $V(t) = B\psi(t)$ as described in section 3.

### C.1.3 CLASSIFIER

The classifier takes as input the concatenated context scalars as a vector. A linear layer is then followed by a softmax activation to output class probabilities.

## D MIT BIH: ADDITIONAL DETAILS

Note that our architecture takes some inspiration for the $H$ that we use in our value function from a github repository[5], although they used tensorflow and we implemented our method in pytorch.

### D.1 VALUE FUNCTION

The value function regresses the output of repeated convolutional and max pool layers on basis functions, where the original time series was the input to these convolutional/max pooling layers. All max pool layers have pool size 2. There are multiple sets of two convolutional layers followed by a max pooling layer. The first set of convolutional layers has 16 filters and filter size 5. The second and third each have 32 filters of size 3. The fourth has one with 32 filters and one with 256, each of size 3. The final output has 256 dimensions of length 23. This is then used as our $H$ matrix in Eqn 6.

### D.2 ENCODER

The encoder takes the original time series as input. It has one hidden layer with a ReLU activation function and 64 hidden units. It outputs the attention density parameters.

### D.3 ATTENTION MECHANISM

The attention mechanism takes the parameters from the encoder and forms an attention density. It then computes

$$c = \mathbb{E}_p[V(T)] \tag{18}$$

for input to the classifier.

### D.4 CLASSIFIER

The classifier has two hidden layers with ReLU activation and outputs class probabilities. Each hidden layer has 64 hidden units.

---

[5] https://github.com/CVxTz/ECG_Heartbeat_Classification

