# OpenReview forum: "Kernel Deformed Exponential Families for Sparse Continuous Attention"
_ICLR.cc/2022/Conference — ICLR 2022 Submitted_

### Official Review · Reviewer_eE87 · 2021-10-29

**Correctness:** 4
**Technical Novelty And Significance:** 2
**Empirical Novelty And Significance:** 2
**Recommendation:** 3
**Confidence:** 2

**Main Review:**

The paper is clearly written. However, the results are rather a straightforward extension of the existing work by Martins et al. (2020; 2021), in which finite-dimensional exponential families and their deformed variants are used to formulate the continuous attention mechanism. In Section 4, some conditions for the finiteness of integration are introduced for the kernel-based model. These results may be important as a fundamental property of kernel-based models. However, this paper does not reveal the significant advantage of kernel-based modeling as an ingredient of continuous attention mechanism. As the author pointed out, the numerical integration required in the proposed method is the problem that should be resolved. The current form of the proposed method is far from practical usage.


Some questions:
- How can one choose the parameter alpha in the deformed exponential family? It would be nice to show a data-dependent method of selecting the deformation parameter.
- To compute the context c, Is a resampling method such as the Metropolis-Hastings algorithm an efficient approach?



**Summary Of The Paper:**

The paper studies the continuous attention mechanism using the kernel exponential family and its deformed variant. This is an extension of existing works based on finite-dimensional exponential families. The authors investigated some theoretical conditions such that the RKHS defines the probability density functions. Numerical experiments showed that the proposed method works efficiently for some datasets.


**Summary Of The Review:**

The proposed model in the paper seems a straightforward extension of existing works by Martins et al. (2020; 2021). Hence, the novelty of the paper is limited.

---

### Official Review · Reviewer_wuA4 · 2021-11-02

**Correctness:** 3
**Technical Novelty And Significance:** 3
**Empirical Novelty And Significance:** 2
**Recommendation:** 5
**Confidence:** 3

**Main Review:**

Overall, I think the subject that the manuscript aims to extend is of high current interest. But I think the authors have focused on an overly dry/technical aspect of the problem. The main theoretical contribution seems to be the statement of conditions under which a kernel exponential distribution exists (i.e., the normalization constant is finite).
I don't have any technical objections to this development.
But an attention mechanism is a means for obtaining improved performance, and I would have expected a clearer demonstration that the theoretical development is worth considering from a practical standpoint (i.e., is computationally light enough to be part of a deep network, and improves performance significantly).

In that respect, I found the experiments to be thin in terms of demonstrating how and why one should consider the proposed alternative over existing ones.
Instead of considering three very short experiments, it would be more useful to focus on one experiment with a clear explanation of the computational load, and the steps taken in order to compute the attention vector.

The improvement obtained by using the proposed mechanism was also not clear to me -- for instance in the IMDB example, the results are close to when the attention mechanism is continuous sparsemax (from previous work). For uWave, the proposed attention mechanism does better than when alternatives are usedm, but the accuracy obtained is around the accuracty reported in the original paper (from 2009, i.e., the pre-deep learning era).

I would suggest the authors to focus more on the experimental section.
A discussion on computational cost is also welcome. The authors do mention the intention of replacing the (undesired, in my opinion) numerical integration, but even with Monte Carlo techniques, wouldn't this be a bottleneck?

**Summary Of The Paper:**

The manuscript considers an extension of the attention mechanism framework developed by others in recent work (by Martins et al.). Specifically this framework allows to break free from the discrete nature of attention that typically consists of a weighted average of a finite set of vectors. This is realized by estimating a probability mass function (PMF) over the finite collection of vectors, and then computing the expected value. The generalization allows one to extend the finite collection to a continuum. This is then handled by using a probability distribution function (pdf) over the collection to compute an expected value. In recent work, various authors considered the probability distribution used in defining the attention mechanism to belong to either a unimodal exponential family, a "deformed" exponential family (deformed versions having possibly finite support), or a mixture of Gaussians. Traditional exponential family comprises pdfs that possess a finite set of sufficient statistics. In contrast, the kernel exponential family comprises pdfs that have essentially infinitely many sufficient statistics through the use of a kernel. The current manuscript proposes to employ a kernel exponential family, and a deformed kernel exponential family. This allows them to work with multimodal probability distributions, and/or distributions with compact support. The authors layout conditions under which the kernel versions of the (deformed) exponential family are defined. They also apply the new attention schemes to several datasets.

**Summary Of The Review:**

I think the manuscript considers a subject of high current interest. The contribution is mostly theoretical, laying out conditions under which a pdf from a family exists (which is a fair question to consider). While I don't have any technical objections, the attention mechanism looks computationally costly, and I wonder if this is going to be an issue during training in practice. Also, I found the experimental evaluation weak, and hand-wavy. I would have expected a clear demonstration of how the method is used, what its computational cost is, and how it improves upon existing alternatives.

---

### Official Review · Reviewer_w8oo · 2021-11-07

**Correctness:** 4
**Technical Novelty And Significance:** 3
**Empirical Novelty And Significance:** 2
**Recommendation:** 6
**Confidence:** 2

**Main Review:**

This work is well-motivated and the authors clearly highlighted differences with previous work (section 3).
Moreover, the contribution is well founded and the authors conducted detailed theoretical analysis of their approach.

However, there are computational limits in the proposed method, as highlighted by the authors: the partition function lacks of a closed form expression (end of page 7).
Therefore, they rely on numerical integration to compute it.
Although this opens up possible research direction for future work, I would like to have more information about experrimental speed efficiency of kernel deformed exponential families  compared to standard sparsemax/softmax.

Moreover, the benefit of the attention mechanism proposed by the authors is that it allows sparsity and multimodality: the paper would benefit of including experimental evidence that this mechanism is important, beside test-set accuracy. What is the actually sparsity ratio compared to sparsemax? How important is the multimodality property? (i.e. are there many instances where the attention map is *really* multimodal? Is it possible to quantify this?)

**Summary Of The Paper:**

Many modern neural architectures, especially in natural language processing, rely heavily on the attention mechanism.
Previous work in the literature proposed to extend the softmax-based attention mechanism by using different distribution families.
In particular, the authors of this paper focus on variants of the attention mechanism that allow for continuous and sparse attention.

The authors propose kernel deformed exponential families, an extension of the exponential family that allows sparse and multimodal attention, contrary to most of previous work that focused on unimodal attention.

**Summary Of The Review:**

Interesting work, however the experiments section lack of qualitative results regarding the proposed approach (i.e. multimodality benefit is only quickly showed for one dataset in Figure 1)

---

### Author Response · Authors · 2021-11-22
**Reviews**

Dear Reviewers,

Thank you for the helpful comments. It appears that reviewers feel that it is not quite ready for publication. We will work to incorporate the comments to submit to the next conference. Particularly we will characterize the rate of convergence of numerical integration in our setting both theoretically and empirically, and go more in depth into at least one of the experiments. Reviewer 1/w800, thank you for the suggestion of characterizing how often sparsity and multimodality show up in attention densities. Reviewer 2/wuA4, your point that 'an attention mechanism is a means for obtaining improved performance, and I would have expected a clearer demonstration that the theoretical development is worth considering from a practical standpoint' gives us a helpful way of guiding our future applications. Reviewer 3/eE87, your point that we need to show the advantage of adding kernels will similarly be a guiding point.

Best,
Authors

---

### Decision · Program_Chairs · 2022-01-20

**Decision:**

Reject

**Comment:**

This paper extends the recent work on continuous-domain sparse attention mechanisms to use kernel parametrizations, and thus allow more flexible multi-modal shapes. Continuous attention extends the standard attention mechanisms to continuous-valued key/value/query functions, involving integrals over probability measures instead of sums over softmax-weighted sums.

Kernel methods fit very well in the framework and provide great expressivity. Reviewers agree it is an interesting and well-motivated idea. The contribution of incorporating kernel families in continuous attention seems substantially novel in comparison to the previous work on the topic.

The main concern, however, is that the paper focuses too much on the theory and not enough on the modeling benefits  enabled by flexible kernels. I would stress that this isn't a question of *improving performance* purely (although quantitative results would help!) but perhaps more of qualitative results, demonstrating e.g. multimodality, selectivity, interpretability.

I very much look forward to a revised version, which I expect would be a strong paper.